# Acetylcholinesterase Inhibition Activity of *Hippeastrum papilio* (Ravenna) Van Scheepen (Amaryllidaceae) Using Zebrafish Brain Homogenates

**DOI:** 10.3390/life13081721

**Published:** 2023-08-10

**Authors:** Luciana R. Tallini, Camila Rockenbach da Silva, Tatiana Jung, Elen de Oliveira Alves, Samira Leila Baldin, Miriam Apel, Luis F. S. M. Timmers, Eduardo Pacheco Rico, Jaume Bastida, José Angelo S. Zuanazzi

**Affiliations:** 1Programa de Pós-Graduação em Ciências Farmacêuticas, Faculdade de Farmácia, Universidade Federal do Rio Grande do Sul, Porto Alegre 90610-000, RS, Brazil; lucianatallini@gmail.com (L.R.T.); elenoliveira11@yahoo.com (E.d.O.A.);; 2Grup de Productes Naturals, Departament de Biologia, Sanitat i Medi Ambient, Facultat de Farmàcia i Ciències de l’Alimentació, Universitat de Barcelona, 08028 Barcelona, Spain; 3Centro de Ciências da Vida, Universidade do Vale do Taquari, Lajeado 95914-014, RS, Brazil; camila.silva5@universo.univates.br (C.R.d.S.); luis.timmers@univates.br (L.F.S.M.T.); 4Programa de Pós-Graduação em Ciências Médicas (PPGCM), Universidade do Vale do Taquari, Lajeado 95914-014, RS, Brazil; tatiana.jung@universo.univates.br; 5Laboratório de Psiquiatria Translacional, Programa de Pós-Graduação em Ciências da Saúde, Universidade do Extremo Sul Catarinense, Criciúma 88806-000, SC, Brazileduprico@gmail.com (E.P.R.); 6Programa de Pós-Graduação em Biotecnologia, Universidade do Vale do Taquari, Lajeado 95914-014, RS, Brazil

**Keywords:** acetylcholinesterase inhibition, Amaryllidaceae, galanthamine, *Hippeastrum papilio*, molecular docking, zebrafish

## Abstract

The Amaryllidaceae family constitutes an interesting source of exclusive alkaloids with a broad spectrum of biological activity. Galanthamine, the most relevant one, has been commercialized for the palliative treatment of Alzheimer’s disease symptoms since 2001 due to its potential as an acetylcholinesterase (AChE) inhibitor. In vitro screenings against AChE by applying different Amaryllidaceae species and alkaloids have been reported in the literature; however, they are usually carried out using purified market enzymes. The main goal of this work is to evaluate the AChE inhibitory potential of *Hippeastrum papilio* (Amaryllidaceae) extracts using zebrafish brain homogenates. The biological assays show that the *H. papilio* bulb extracts present an interesting AChE inhibitory activity in comparison with the positive reference control galanthamine (IC_50_ values of 1.20 ± 0.10 and 0.79 ± 0.15 μg/mL, respectively). The chemical profile of *H. papilio* shows that this species has a high amount of galanthamine, which may contribute to the inhibitory effect on AChE activity of zebrafish brains. Computational experiments were used to build the model for zebrafish AChE and to evaluate the interactions between galanthamine and the enzymic active site. This work suggests that zebrafish could represent an important model in the search for bioactive molecules from the Amaryllidaceae family for the treatment of Alzheimer’s disease.

## 1. Introduction

The search for medicinal plants has always been of paramount importance for human health and wellbeing [1]. Natural products serve as a significant reservoir of active compounds possessing diverse medicinal properties, making them invaluable in the field of medicine [2]. The isolation and characterization of novel molecules, coupled with an understanding of their biological activities, contribute significantly to advancements in human health [3]. It is noteworthy that over 50% of the new approved drugs from 1981 to 2019 were directly or indirectly derived from natural products, underscoring their pivotal role in drug development [2] and highlighting the immense potential and enduring relevance of natural products in shaping the landscape of modern healthcare.

Alkaloids, classified as secondary metabolites, are predominantly encountered in the plant kingdom. These compounds are distinguished by the presence of at least one nitrogen atom within a heterocyclic ring and exhibit an extensive range of structural diversity coupled with a broad spectrum of biological potential [4,5]. Notably, a fraction of these intriguing molecules possess the capacity to engage with the central nervous system (CNS), thereby conferring upon them profound medical and socioeconomic significance [6,7]. The ability of alkaloids to interact with the CNS underscores their pivotal role in various therapeutic interventions and highlights their impact on both individual wellbeing and larger societal dimensions [6,7].

Amaryllidoideae, a subfamily group of the Amaryllidaceae plant family, presents an exclusive and still expanding group of alkaloids known as the Amaryllidaceae alkaloids [8,9]. These compounds have an interesting structural diversity and a broad biological potential, such as antiprotozoal, antiviral, antitumoral, anti-inflammatory, anticholinesterase, and other pharmacological activities [8]. Moreover, the Amaryllidaceae alkaloids exhibit a wide range of therapeutic effects, making them highly significant in drug discovery and development. Usually, the Amaryllidaceae alkaloids can be classified into nine main skeleton types, named norbelladine, lycorine, homolycorine, crinine, haemanthamine, narciclasine, tazettine, montanine, and galanthamine. However, a growing body of scientific literature has documented the discovery of numerous additional skeleton types, bringing the total number to 42 distinct skeletal structures [8,10].

Acetylcholinesterase (AChE) is a highly selective cholinergic enzyme involved in the hydrolysis of the acetylcholine (ACh) neurotransmitter that plays an important role in maintaining the cortical activity and cerebral blood flow, modulating cognition, learning, task and memory-related activities, the development of the cerebral cortex, and regulation of the sleep–wake cycle [11,12]. In 2001, the Food and Drug Administration (FDA) approved the use and commercialization of galanthamine, an Amaryllidaceae alkaloid, under its salt form for the palliative treatment of mild to moderate Alzheimer’s disease symptoms due to its potential as an AChE inhibitor [13]. This fact attracted the attention of researchers to the Amaryllidaceae family, which had the potential to be an interesting source for new bioactive molecules [14]. Biological screenings using alkaloid extracts and/or purified alkaloids from this plant family have been carried out for the last few decades. Consequently, the search for different methodologies surrounding the anticholinesterase potential of these structures may contribute to a better understanding of the Amarylliadaceae alkaloids and their potential in the development of new drugs for Alzheimer’s disease.

Zebrafish (*Danio rerio*) is a tropical freshwater fish native to Southeast Asia that has become an important organism for different areas of biological research, including neuroscience [15,16,17,18]. This species presents a short development time to sexual maturity (3 months) and a high reproductive rate [19]. Furthermore, it appears to provide a good correlation between system complexity and practical simplicity [20]. Additionally, zebrafish as a model related to human pathogenesis has an increasing importance for screening in drug discovery due its genetic makeup; it shares a homology of 70–80% with humans [21,22]. Zebrafish models are an interesting tool that could be strategically incorporated into analysis of the neurodegeneration cascade, covering the prevailing gaps between the drug discovery in vitro models and the preclinical assays in rodents; zebrafish also seem to be attractive alternative models for Alzheimer’s disease research [23].

The aim of this study was to evaluate the cholinergic inhibitory activity of *Hippeastrum papilio* (Ravenna) Van Scheepen (Amaryllidaceae) extract using zebrafish as a model. The alkaloid profile of this plant was analyzed by gas chromatography coupled to mass spectrometry (GC-MS). The inhibitory potential of *H. papilio* alkaloid extract against AChE was evaluated in vitro using zebrafish homogenized brain extract instead of the purified market enzyme. In silico experiments were carried out to better understand the interactions between galanthamine and the active site of zebrafish AChE.

## 2. Materials and Methods

### 2.1. Plant Material

Bulbs of *Hippeastrum papilio* (Ravenna) Van Scheepen, belonging to the Amaryllidaceae family, were collected in Porto Alegre, situated in the state of Rio Grande do Sul, Brazil, during the month of April in the year 2019. Prior to this collection, the plant had undergone comprehensive identification by Dr. Julie H. A. Dutilh from the University of Campinas (Unicamp), located in Campinas, Brazil. A corresponding specimen voucher (ICN-149428) was subsequently deposited at the Institute of Botany, which is affiliated with the Federal University of Rio Grande do Sul (UFRGS) in Porto Alegre, Brazil.

### 2.2. Extraction

A quantity of 0.5 kg of fresh bulbs from *Hippeastrum papilio* was subjected to a drying process at a controlled temperature of 40 °C for seven days. Subsequently, the resulting dried material, weighing 62.8 g, was triturated and underwent a maceration procedure. The maceration process was carried out with methanol (2 × 500 mL) and conducted at room temperature for a total of 48 h. Following the completion of the maceration, the combined macerate was subjected to filtration, facilitating the separation of solid particulates. The solvent was subsequently evaporated under reduced pressure until complete dryness was achieved. Upon reaching this stage, the resultant crude extract, weighing 20 g, was subjected to acidification by the addition of diluted sulfuric acid (2%, *v*/*v*) until a pH of 3 was obtained. To further clean the extract, the neutral material was selectively removed using diethyl ether in five successive extractions, each employing 100 mL of the solvent. The remaining aqueous solution was then subjected to basification by the addition of ammonium (25%, *v*/*v*) until a pH of 10 was achieved. This alkaline solution was then extracted with ethyl acetate in five sequential extractions, each one utilizing 100 mL of the solvent. The cumulative effect of these extractions resulted in the obtainment of the alkaloid extract, which weighed 303.6 mg.

### 2.3. GC-MS Analysis

A total of 2 mg of the alkaloid extract was dissolved in 1 mL of methanol and chloroform (1:1, *v*/*v*), and 1 μL of the sample was injected, using manual mode, into the GC-MS apparatus (Agilent Technologies, Santa Clara, CA, USA, 5975C coupled with 7890A) operating in the electron ionization (EI) mode at 70 eV. A Durabond-DB-5 MS column (John Wiley & Sons Scientific, Hoboken, NJ, USA, 30 m × 0.25 mm × 0.25 μm) was used. The chromatographic conditions were adapted from Berkov and co-authors [24]. The temperature gradient performed was as follows: 12 min at 100 °C, 100–180 °C at 15 °C/min, 180–300 °C at 5 °C/min, and 10 min hold at 300 °C. The injector and detector temperatures were 250 and 280 °C, respectively, and the flowrate of carrier gas (He) was 1 mL/min.

### 2.4. Alkaloid Identification

The alkaloids present in the extract derived from *Hippeastrum papilio* bulbs were identified through a comprehensive analysis involving the comparison of their gas chromatography–mass spectrometry (GC-MS) spectra with those of authenticated Amaryllidaceae alkaloids. These authentic alkaloids had been previously isolated and identified using a range of spectrometric techniques such as nuclear magnetic resonance (NMR), ultraviolet (UV) spectroscopy, circular dichroism (CD), infrared (IR) spectroscopy, and mass spectrometry (MS). This rigorous identification process was conducted by the Laboratory of Natural Products at the University of Barcelona, Spain. In addition, a combination of the NIST 05 Database and relevant data from literature sources was also employed.

### 2.5. Alkaloid Quantification

To quantify the single constituents, a calibration curve of galanthamine (100, 200, 300, 500, and 600 μg/mL) was used. The peaks areas were manually obtained considering selected ions for each compound (generally the base peak of their MS). The ratio between the values obtained for galanthamine in each solution was plotted against the corresponding concentration of galanthamine to obtain the calibration curve and its equation (y = 22,570x − 662,859; R^2^ = 0.9899).

### 2.6. Determination of AChE Activity

In essence, the AChE inhibition assay was based on the Ellman method [25] with modifications [26]; however, commercial enzymes were not used herein. In order to achieve a closer approximation to an in vivo environment, whole brains of zebrafish, which naturally contain the AChE protein, were used as an enzymatic solution. Adult short-fin wild-type zebrafish (*Danio rerio*) of both sexes (5 months old, weighing 0.400 ± 0.05 g), obtained from the department of Biochemistry at the Federal University of Rio Grande do Sul (UFRGS), were used. The water was kept at 26 ± 2 °C under a 14 h/10 h light–dark cycle photoperiod and fed twice a day with artemia and commercial flake fish food. All procedures presented in this study were performed in agreement with the National Institute of Health Guide for Care and Use of Laboratory Animals and approved by the local ethics committee. Ten whole brains were removed by dissection for each sample and homogenized in a glass–Teflon homogenizer on ice in 1 mL of Tris-citrate buffer (50 mM Tris, 2 mM EDTA, and 2 mM EGTA, pH 7.4, adjusted with citric acid). Protein was measured using Coomassie blue as the color reagent [27] and bovine serum albumin as the standard. After measuring the concentration, the volume in the zebrafish brain extract was normalized and an amount of 10 μg was determined.

The AChE inhibitory activity of the alkaloid extracts was evaluated in a 96-well microplate, and phosphate buffer (8 mM K_2_HPO_4_, 2.3 mM NaH_2_PO_4_, 0.15 M NaCl, pH 7.5) was used in this assay. An amount of 2 μL of zebrafish brain protein and 48 μL of phosphate buffer were added to the plate. Then, 50 μL of the Amaryllidaceae alkaloid extract was added and the plate incubated for 30 min at room temperature. Following this, 100 μL of the substrate solution (0.04 M Na_2_HPO_4_, 0.2 mM DTNB, and 0.24 mM ACTI in water, pH 7.5) was added. After 10 min, the absorbance of the reaction was read in a spectrophotometer at 405 nm. The enzyme activity was calculated as a percentage compared with the control using a buffer without any inhibitor. Galanthamine alkaloid was used as a reference control. The AChE inhibition activity of the *H. papilio* bulb alkaloid extract was calculated using a calibration curve (0.5, 1, 2, 3, 4, 5, and 10 μg/mL) to determine the IC_50_ value of the plant sample (using the software Prism 9).

### 2.7. Computational Experiments

#### 2.7.1. Comparative Modeling

The homology modeling approach, implemented in the MODELLER 9v23 program [28], was used to build the model for *Danio rerio* AChE (Uniprot: Q9DDE3) based on the three-dimensional structure of AChE from *Tetronarce californica* (PDB code: 1GQR; 66% sequence identity) associated with rivastigmine. The protocol used to perform the comparative modeling experiments involved the generation of five models. All models were submitted to the evaluation of the energy function DOPE [29], aiming to choose the best structures. In addition, we used the MOLPROBITY webserver [30] and the PROCHECK program [31] to verify the stereochemical quality of the models.

#### 2.7.2. Molecular Docking Protocol

Molecular docking simulation was carried out to analyze the orientation and binding affinity of galanthamine into the binding site of *Danio rerio* AChE. The flexible docking simulations were performed using PyrX-0.9.8 [32], where the AutoDock Vina is implemented [33]. Initially, a re-docking simulation was performed using the PDB ID 1GQR to verify the accuracy and precision of the docking program to find the same ligand’s position observed in the crystallographic structure. A grid box was created with 15 × 15 × 15 angstroms, centered on the coordinates x = −5.28, y = 80.28, and z = 66.88, in order to include solely the protein’s active site to reduce the computational cost. The molecular docking process was carried out with 10 independent runs and the exhaustiveness parameter was set to 30.

### 2.8. Ethical and Legal Considerations

The project was approved by the Ethics Committee on Animal Use (CEUA/UFRGS), number 36120. The access authorization and collection for the plant species was emitted by the CGEN under AA621EE.

## 3. Results and Discussion

### 3.1. Chemical Profile

A total of seven distinct structures were successfully detected and identified within the alkaloid extract derived from *Hippeastrum papilio* utilizing gas chromatography–mass spectrometry (GC-MS). To determine the concentration of each individual constituent present in the chromatogram, a calibration curve for galanthamine was employed. The resulting quantification values were reported in micrograms of galanthamine per milliliter (µg GAL/mL). It is important to note that the quantification obtained through this methodology is not absolute; however, it provides a reliable means of comparing the relative amounts of specific alkaloids within Amaryllidaceae samples [34].

In addition to quantification, the percentage composition of each constituent in the sample was calculated, shedding light on the relative distribution of these alkaloids within the *Hippeastrum papilio* extract. Detailed information regarding these quantification and percentage values can be found in Table 1, whereas Figure 1 shows the identified alkaloid structures found in *H. papilio* and in Figure 2 it is possible to find chromatograms.

The alkaloid profile obtained herein fits with other results for Brazilian *H. papilio* samples. De Andrade and co-authors [35] described the presence of galanthamine, narwedine, 11β-hydroxygalanthamine, haemanthamine, 8-*O*-demethylmaritidine, vittatine, and 11-hydroxyvitattine in this species. Later, Guo and co-workers [36] also reported the occurrence of galanthamine, narwedine, 11β-hydroxygalanthamine, and haemanthamine (as in the previous investigation [35]), along with apogalanthamine, hippapiline, papiline, 9-*O*-demethyllycosinine B, and 3-*O*-demethyl-3-*O*-(3′-hydroxybutanoyl)haemanthamine, which were identified in *H. papilio* for the first time [36]. Although variabilities in the alkaloid profile are noticed in both studies, galanthamine remained the main compound identified in these extracts. A patent also reported the high occurrence of galanthamine in a cultivated *H. papilio* sample [37], and differences in the alkaloid profiling of Amaryllidaceae species have also been observed among *Lapiedra martinezii* species collected in different regions of Spain [38], as well as among *Rhodophiala andicola* species collected in different Chilean localities [39].

As shown in Table 1, the main compound in *H. papilio*, galanthamine (**1**), was determined herein at a 1286.03 μg GAL/mL concentration, which corresponds to 67.3% of the total identified alkaloid in this species. According to the literature, this alkaloid was firstly isolated from *Galanthus woronowii* in 1952, and nowadays, it is still obtained by pharmaceutical companies from natural sources of Amaryllidaceae plants, such as *Galanthus nivalis*, *Leucojum aestivum*, *Lycoris radiata,* and different species of *Narcissus* [40]. Narwedine (**3**) and 11β-hydroxygalanthamine (**6**), both belonging to galanthamine-type alkaloids, were also identified in *H. papilio* samples; however, these compounds were in lower quantities than galanthamine (**1**), see Table 1. According to the literature, the compound (**3**) has been detected in different members of the Amaryllidaceae genus, such as *Chlidanthus*, *Galanthus*, *Hippeastrum*, *Narcissus,* and *Pancratium* [41,42,43,44,45], whereas compound (**6**) has been reported in *H. papilio* samples [35,36].

The compounds papiline (**2**) and 9-*O*-demethyllycosinine B (**5**) identified here (Table 1) are galanthindole-type structures and were described for the first time in *H. papilio* by Guo and co-authors [36] and in *H. breviflorum* by Sebben and co-authors [46]; both plants were collected in Brazil. The galanthindole-type alkaloids are derived from homolycorine-type structures [10]. Hippapiline (**4**) is a homolycorine-type structure that is reported in Table 1 and Figure 1.

### 3.2. AChE Inhibitory Activity

Acetylcholine (ACh) undergoes hydrolysis through the action of two distinct cholinesterases: acetylcholinesterase (AChE) and butyrylcholinesterase (BuChE) [47]. The existing literature demonstrates that BuChE is absent from the genome of *Danio rerio*, commonly known as zebrafish, whereas AChE is encoded by a single gene that has been successfully cloned, sequenced, and functionally identified within the zebrafish brain [48].

To assess the potential inhibitory effects of the alkaloid extract from *Hippeastrum papilio* on AChE, it was deemed important to replicate the physiological conditions. Consequently, homogenates of zebrafish brains, which inherently contain AChE, were employed as the enzymatic solution instead of purified AChE enzyme. This approach aimed to approximate the ideal conditions of a physiological medium for the evaluation of AChE inhibition exerted by the alkaloid extract from *H. papilio*. The utilization of a biologically derived enzymatic system further enhances the translational value of the findings, facilitating the extrapolation of results to potential physiological implications.

The alkaloid extract of *H. papilio* presented a very high AChE inhibition activity, with IC_50_ values of 1.20 ± 0.10 µg/mL, whereas galanthamine (**1**), which was used as a positive control, showed IC_50_ values of 0.79 ± 0.15 µg/mL (see Figure 3). The alkaloid composition is an essential point for understanding the biological properties of Amaryllidaceae species. The chemical profile obtained by GC-MS indicated the existence of galanthamine-type structures in the *H. papilio* sample, such as 67.3% of galanthamine (**1**), 3.2% of narwedine (**3**), and 5.7% of 11β-hydroxygalanthamine (**6**), which suggested the potential of this plant extract as a great AChE inhibitor (Figure 3).

Applying a similar methodology, Pitchai and co-authors [49] evaluated the AChE inhibitory potential of trans-tephrostachin (a compound obtained from *Tephrosia purpurea* extract) in zebrafish brain homogenate and in human AChE. The study showed IC_50_ values of 39.0 ± 1.4 for the former and 20.0 ± 1.0 µM for the latter. The authors also used galanthamine as positive control and obtained IC_50_ values of 3.3 ± 0.5 µM in the zebrafish model [49].

Previous investigations have described that when using the classical methodology of Ellman and co-workers [25] and López and co-workers [26], the IC_50_ galanthamine value necessary for activity against purified *Electrophorus electricus* AChE (eeAChE) is about 0.50 μg/mL, which correspond to approximately 1.7 μM [39]. Some authors reported that narwedine (**3**) and 11β-hydroxygalanthamine (**6**) show inhibitory activity against *ee*AChE, with IC_50_ values of 9.38 and 3.49 µM, respectively [45]. Furthermore, researchers have shown that habranthine (an epimer of 11β-hydroxygalanthamine, described as 11-hydroxygalanthamine by the authors) exhibits similar eeAChE inhibitory activity to galanthamine, with IC_50_ values of 1.61 and 1.07 μM, respectively [26].

According to the literature, different extracts of Amaryllidaceae species have presented anti-cholinesterase potential, which is usually correlated with the presence of galanthamine-type alkaloids in these samples [26]. Researchers have recently shown the potential of *Hieronymiella marginata* (Amaryllidaceae) extract against *ee*AChE, suggesting that its inhibitory activity (with IC_50_ values of 1.84 ± 0.01 μg/mL) could be related to the presence of sanguinine in this sample; galanthamine, used as control, showed IC_50_ values of 1 ± 0.05 μM [50]. The alkaloid sanguinine is considered more potent than galanthamine against the AChE enzyme; however, galanthamine has a better ability to cross the blood–brain barrier than sanguinine [51].

### 3.3. Computational Results

According to the available literature, galanthamine exerts its effects on human acetylcholinesterase (AChE) in a dual manner, functioning as both a competitive and reversible inhibitor [52,53]. Additionally, galanthamine has been identified as an allosteric ligand of acetylcholine (Ach) nicotinic receptors (nAChR) in humans [52,53]. In a recent study conducted by Pitchai et al. [49], it was demonstrated that galanthamine displays competitive inhibition of AchE in the zebrafish brain, resulting in increased Km values while exhibiting a minimal impact on the maximum velocity of the enzyme activity.

To gain deeper insights into the interactions between galanthamine and the AChE enzyme within zebrafish brain homogenate, computational experiments were conducted. Consequently, a computational model of the AChE enzyme from *Danio rerio* (Uniprot: Q9DDE3) was constructed. This model serves as a valuable tool for simulating and investigating the specific binding interactions and molecular mechanisms underlying the galanthamine–AChE interaction within the zebrafish brain.

According to the docking results, galanthamine binds to the zebrafish AChE active site with an estimated free binding energy of −9.3 kcal/mol (see Table 2). We observed that this compound maintains hydrogen bond interactions with the side chain of Tyr119 and two more hydrophobic interactions with the Trp81 and Tyr328 residues. The conformation of the galanthamine into the *D. rerio* AChE active site is shown in Figure 4, and Figure 5 illustrates the interactions observed between the other alkaloids identified in *H. papilio* and the active site of DrAChE. The molecular docking of galanthamine on human AChE (PDB: 4EY7) shows that galanthamine presents a hydrogen bond with Ser203 interaction that plays an important role in this enzymic active site [54,55]. The aromatic residues Trp286, Tyr124, Tyr72, and Tyr341 in the human AChE peripheral site form π-cation interactions with ACh and orient it to slide down to the Trp86 and Tyr337 of the choline-binding pocket in the acylation site that is aligned with the catalytic serine [55].

## 4. Conclusions

In summary, substantial efforts have been dedicated to exploring the acetylcholinesterase (AChE) inhibitory potential of Amaryllidaceae species in recent years by employing purified enzymes from various organisms. In this study, we aimed to replicate the ideal conditions of a physiological medium by utilizing zebrafish brain homogenate as a substitute for the purified AChE enzyme. The alkaloid extract obtained from *Hippeastrum papilio*, which demonstrated significant galanthamine content by GC-MS, exhibited a notable inhibitory effect in zebrafish brain homogenates.

Through the implementation of a computational model for zebrafish AChE, a deeper understanding of the interactions between galanthamine and the amino acid residues within the active site of this enzyme was attained. The insights gained from this computational analysis provided valuable information about the specific molecular interactions underlying the inhibitory activity of galanthamine in zebrafish AChE.

These findings contribute to the broader understanding of Amaryllidaceae alkaloids and their interactions with AChE, highlighting the potential utility of zebrafish as an effective model for the preliminary evaluation of acetylcholinesterase inhibitory activity and paving the way for further investigations and subsequent preclinical studies.

## Figures and Tables

**Figure 1 life-13-01721-f001:**
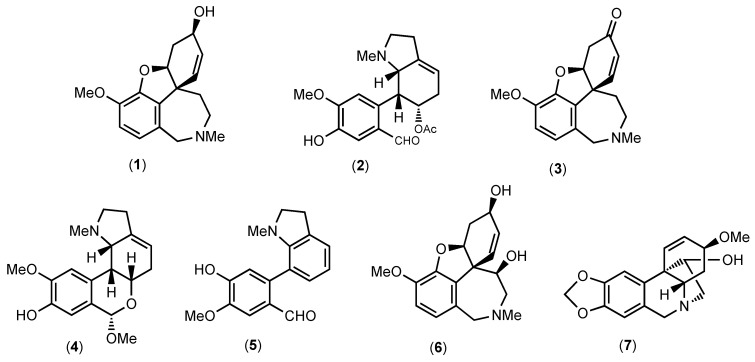
Structures of alkaloids identified in *Hippeastrum papilio* extract by GC-MS.

**Figure 2 life-13-01721-f002:**
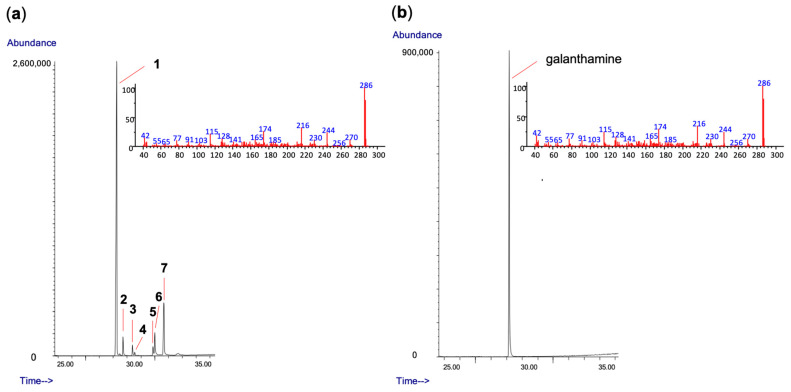
(**a**) GC chromatogram of the alkaloid extract of *Hippeastrum papilio* along with the MS of the galanthamine peak; (**b**) GC chromatogram of galanthamine (reference compound) accompanied by its MS spectrum.

**Figure 3 life-13-01721-f003:**
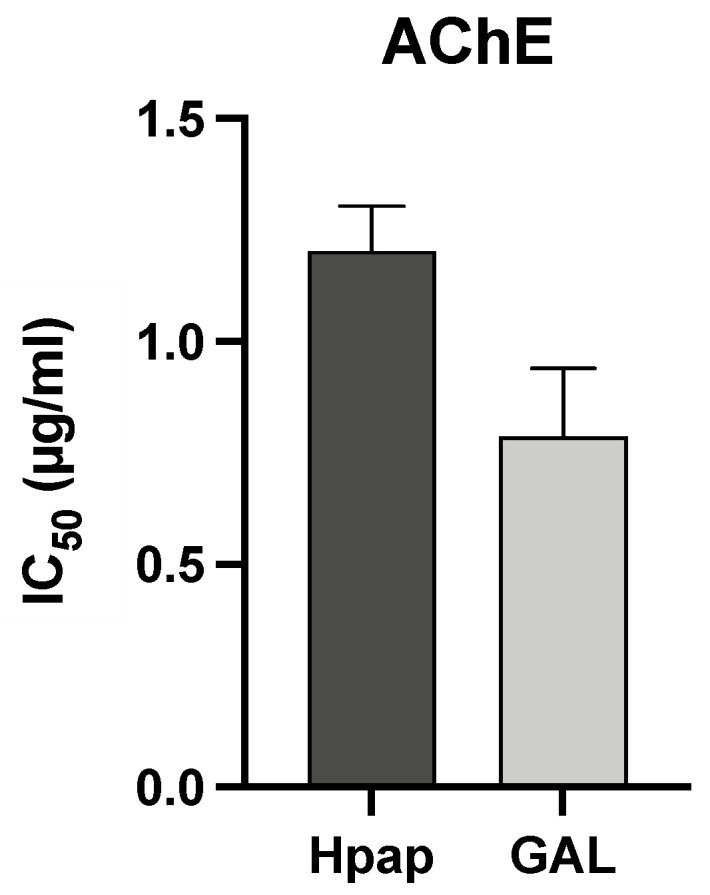
Inhibitory activity of *Hippeastrum papilio* alkaloid extracts against AChE by applying zebrafish brain homogenate. Values are expressed as IC_50_. The data presented are the mean values plus SD (n = 3). PRISM software was used. Hpap = *Hippeastrum papilio*; GAL = galanthamine (reference compound).

**Figure 4 life-13-01721-f004:**
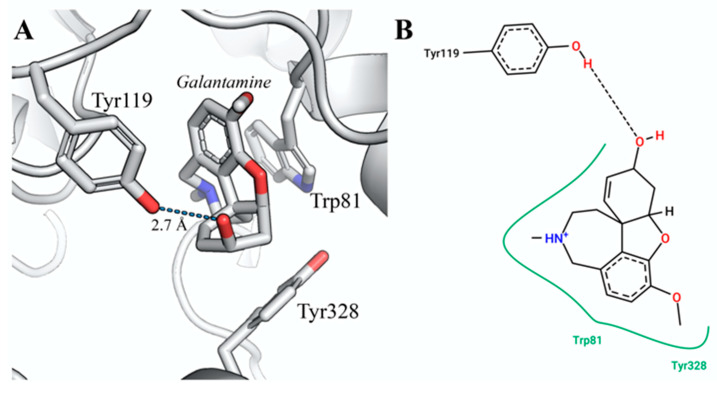
The best docking conformation of the complex DrAChE:galanthamine interaction. (**A**) The active site of DrAChE associated with galanthamine. The overall structure is represented as a cartoon. The galanthamine and residue sidechains are shown as sticks and colored in CPK. (**B**) Schematic diagram of the protein–ligand interactions. The dashed line shows the hydrogen bond, whereas the green line represents the residues involved in hydrophobic contacts. Image generated with PyMOL (PyMOL Molecular Graphics System, Version 1.5.0.4 Schrödinger, LLC.) and PoseView [56,57].

**Figure 5 life-13-01721-f005:**
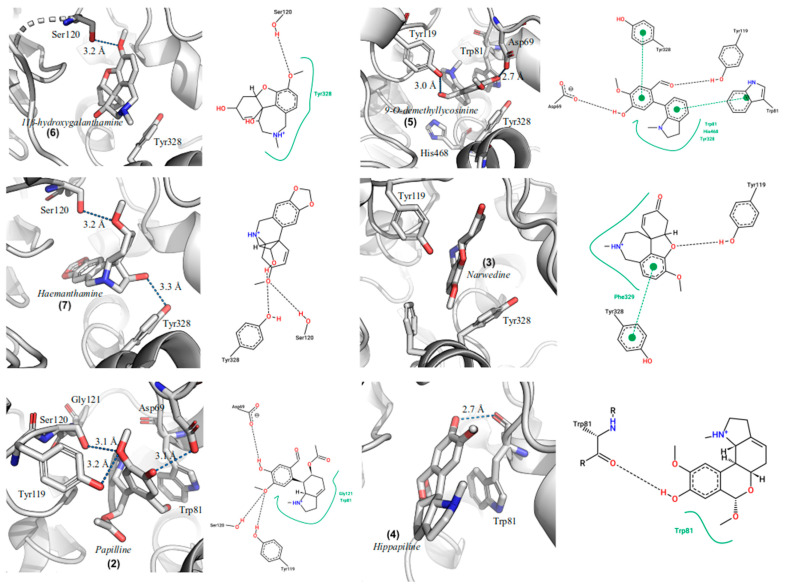
Graphical representations of the best docking conformation of the binding of (**2**) papiline, (**3**) narwedine, (**4**) hippapiline, (**5**) 9-*O*-demethyllycosinine, (**6**) 11β-hydroxygalanthamine, and (**7**) haemanthamine in the gorge of the active site of DrAChE.

**Table 1 life-13-01721-t001:** Chemical profile of *Hippeastrum papilio* alkaloid extracted by GC-MS.

Alkaloid	Rt	Fragmentation Pattern, *m/z* (Relative Intensity, %)	Amount ^a^	% ^b^
galanthamine (**1**)	27.247	287 (78), 286 (100), 270 (12), 244 (23), 216 (32), 174 (26)	1286.03	67.3
papiline (**2**)	27.715	286 (3), 177 (16), 110 (8), 109 (100), 108 (24), 94 (5)	89.77	4.7
narwedine (**3**)	28.395	285 (80), 284 (100), 242 (19), 216 (21), 199 (22), 174 (31)	61.31	3.2
hippapiline (**4**)	28.564	177 (15), 110 (9), 109 (100), 108 (23), 94 (7), 81 (5)	38.25	2.0
9-*O*-demethyllycosinine B (**5**)	29.881	283 (100), 255 (75), 254 (83), 240 (31), 222 (39), 194 (20)	56.95	2.9
11β-hydroxygalanthamine (**6**)	30.009	303 (24), 286 (10), 231 (22), 230 (100), 213 (34), 197 (10)	106.57	5.7
haemanthamine (**7**)	30.657	301 (13), 273 (18), 272 (100), 240 (19), 181 (26), 153 (14)	271.78	14.2
**Total**	**-**	**-**	**1910.66**	**100.0**

Rt: retention time; ^a^ Values are expressed as μg GAL/mL; ^b^ Values expressed as percentage of amount (μg GAL/mL).

**Table 2 life-13-01721-t002:** Estimated free binding of molecular docking between alkaloids identified in *Hippeastrum papilio* and zebrafish AChE. Values expressed in kcal/mol.

Alkaloid	Estimated Free Binding Energy
galanthamine (**1**)	−9.3
papiline (**2**)	−9.1
narwedine (**3**)	−9.5
hippapiline (**4**)	−9.6
9-*O*-demethyllycosinine B (**5**)	−9.0
11β-hydroxygalanthamine (**6**)	−9.7
haemanthamine (**7**)	−8.6

## Data Availability

Data is contained within the article.

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
