# Peer review of "Acetylcholinesterase Inhibition Activity of Hippeastrum papilio (Ravenna) Van Scheepen (Amaryllidaceae) Using Zebrafish Brain Homogenates"

_life, 2023, doi:10.3390/life13081721_

Round 1
Reviewer 1 Report
The manuscript discussed anti-AChE activity of Hippeastrum papilio (Amaryllidaceae) extract using zebrafish brain homogenates extract beside chemical profile of alkaloids using GC/MS and Molecular docking.
There are some comments should be done:
1. The reference of GC/MS method should be cited.
2. The computational studies should be done for all identified alkaloids.
2. What is the novelty of the manuscript?
The manuscript language should be revised.
Reviewer 2 Report
In this paper, the authors reported a new method of testing the AChE inhibitory activity of H. papilio extract by using zebrafish brain homogenates extract, which can kind of mimic the in vivo activity. The paper is well-designed and written; however, minor revision is required before acceptance.
1. The authors use zebrafish brain homogenates extract to evaluate the AChE inhibitory activity; how do you control the concentration of the enzyme?
2. In Figure 1, please re-draw the structures in a uniform style.
3. In Figure 2, compound 4 didn’t clearly show. There are two extra peaks, one between 1 and 2, the other one after 7, what are they? The quality of the figure should be improved.
4. Figure 2, since compound 1 is the main compound, and maybe also the main active one, the author should add the GC standard of 1.
5. Figure 3, please make it clear y-axis is IC50 concentration.
6. All the compound numbers should be bolded.
7. Part 2.6, some clerical errors, “AChE protein, ware used as enzymatic”; “Amaryllidaceae alkaloid extract were added, and he plate was”.
Minor editing of English language required
